# Beyond the Laplacian: Interpolated Spectral Augmentation for Graph Neural Networks

## Abstract

Graph neural networks (GNNs) are fundamental tools in graph machine learning. The performance of GNNs relies crucially on the availability of informative node features, which can be limited or absent in real-life datasets and applications. A natural remedy is to augment the node features with embeddings computed from eigenvectors of the graph Laplacian matrix. While it is natural to default to Laplacian spectral embeddings, which capture meaningful graph connectivity information, we ask whether spectral embeddings from alternative graph matrices can also provide useful representations for learning. We introduce Interpolated Laplacian Embeddings (ILEs), which are derived from a simple yet expressive family of graph matrices. Using tools from spectral graph theory, we offer a straightforward interpretation of the structural information that ILEs capture. We demonstrate through simulations and experiments on real-world datasets that feature augmentation via ILEs can improve performance across commonly used GNN architectures. Our work offers a straightforward and practical approach that broadens the practitioner's spectral augmentation toolkit when node features are limited.

## 1 Introduction

Graph neural networks (GNNs) are foundational tools for modeling and learning from relational data. GNNs operate by passing, transforming, and aggregating messages between neighboring nodes iteratively, thereby encoding connectivity and feature information into expressive node representations that can capture complex patterns and relationships. The GNN approach has seen significant success, enabling advances in domains ranging from biology (Stokes et al., 2020) to recommender systems (Ying et al., 2018; Wu et al., 2022). The effectiveness of GNNs, however, often hinges on the availability of rich and informative node-level features. In many practical settings, node features can be unavailable (due to reasons such as privacy, missing data, etc), limited, or corrupted with noise (Rossi et al., 2022). Naïve application of GNN methods in these settings can lead to degraded performance (Said et al., 2023).

In these settings where node features are limited or unavailable, the most natural approach is to perform feature augmentation. A common heuristic is to augment the node features with one-hot encoding vectors or the all-ones vector. Such features do not take into account the underlying graph's topology. A more informative approach is spectral augmentation, where spectral embeddings derived from the graph Laplacian matrix's eigenvectors are used as node features (Said et al., 2023). Indeed, it is well known from the manifold learning and spectral clustering literature that Laplacian spectral embeddings capture rich structural properties of the underlying graph (Ng et al., 2001; Von Luxburg, 2007; Belkin & Niyogi, 2003). This approach of pre-computing spectral embeddings as node features is natural, as it supplies prior structural information that even shallow GNNs can directly exploit, mitigating the need for expensive and deep architectures that often suffer from issues such as oversmoothing (Rusch et al., 2023). Such embeddings have also been successfully applied in GNNs beyond the missing/limited features setting, such as in graph transformers (Kreuzer et al., 2021; Dwivedi & Bresson, 2020) as a form of positional encoding. Lim et al. (2022) showed that such Laplacian embeddings, after accounting for sign and basis symmetries, are expressive and can improve performance of GNNs.

Laplacian spectral embeddings are often treated as the default option in GNN feature augmentation (Said et al., 2023; Lim et al., 2022). On the other hand, a variety of graph matrices motivated by

different applications have been considered in the spectral graph theory and network science literature (Grindrod et al., 2018; Ou et al., 2016; Haemers & Omidi, 2011). These alternative matrices can induce spectral embeddings that capture different kinds of graph connectivity information. For example, Priebe et al. (2019) showed that Laplacian and adjacency spectral embeddings capture drastically different structures of graphs that has substantial implications for downstream tasks such as clustering.

Motivated by the success of spectral feature augmentation in GNNs, especially in the limited/missing node features setting (Said et al., 2023; Lim et al., 2022), we ask whether considering alternative spectral embeddings can lead to improved performance in practice. To this end, we propose a simple yet expressive family of graph matrices that contains many well known graph matrices as special cases. We call the resulting spectral embeddings captured by this family the interpolated Laplacian Embeddings (ILEs). Using tools from spectral graph theory, we provide interpretations for the information captured by ILEs. Through experiments, we show that the choice of graph matrix used to compute spectral embeddings can impact downstream graph learning performance in the missing/limited node features setting. We discuss how to select relevant matrices and tuning parameters in practice. Our results suggest that practitioners many benefit from considering a broader range of spectral embeddings when augmenting node features in GNNs.

## 1.1 RELEVANT LITERATURE

We refer to Hamilton et al. (2018); Wu et al. (2022) for general background on graph neural networks. Commonly used graph neural network architectures that we adopt in our experiments include Graph Convolutional Networks (GCN) (Kipf & Welling, 2017), Graph Isomorphism Networks (GIN) (Xu et al., 2019) and GraphSAGE (Hamilton et al., 2018).

The idea of using Laplacian embeddings as positional encodings to augment input features for graph neural networks has been explored in the literature (Said et al., 2023; Dwivedi et al., 2021). Such Laplacian positional encodings are also used in graph transformers (Dwivedi & Bresson, 2020; Kreuzer et al., 2021). Lim et al. (2022) proposes SignNet and BasisNet as architectural components that can be used to process eigenvector embeddings to ensure invariance to symmetries that arILE from sign flips and rotations (when repeated eigenvalues are present).

In addition to the Laplacian, adjacency spectral embeddings have been explored in the spectral clustering literature (Sussman et al., 2012; Cape et al., 2019). General families of graph matrices and operators have been considered in the spectral graph theory and network science literature. Notable examples include the Katz similarity matrix (Ou et al., 2016), deformed Laplacian (Grindrod et al., 2018), the signless Laplacian (Cvetković et al., 2007) and related generalizations (Nikiforov, 2017), the universal Adjacency matrices (Haemers & Omidi, 2011), and many more. Different spectral embeddings can capture different graph structures with different interpretation, an observation noted in Priebe et al. (2019) in the context of spectral clustering.

**Our Contribution** In this work, we revisit the common practice of using Laplacian spectral embeddings for feature augmentation in graph neural networks. Our contributions are threefold: (i) we introduce interpolated Laplacian Embeddings (ILEs), which are eigenvectors derived from a simple and flexible family of graph matrices that generalizes many classical graph matrices as special cases; (ii) we provide theoretical analysis, leveraging tools from spectral graph theory, to characterize and interpret the structural information that ILEs encode; and (iii) we conduct extensive experiments demonstrating that the choice of graph matrix can lead to improvements in downstream GNN performance, especially under settings with limited or missing node features. Our results move beyond the default use of Laplacian embeddings, broadening the spectral augmentation toolkit available for GNNs.

## 2 PRELIMINARIES

### 2.1 SETUP AND NOTATION

We work in the setting of simple, undirected, connected, weighted graphs. Let $G = (V, E, w)$ denote such a graph, where $V$ is the vertex set, $E$ the edge set, and $w : E \to \mathbb{R}_{\geq 0}$ assigns a

nonnegative weight to each edge. For convenience, we write $w_e$ or $w_{uv}$ to denote the weight of edge $e$ or edge $(u, v) \in E$. Throughout, we index vertices by $[n] := \{1, 2, \ldots, n\}$. We use $i, j, u, v$ to index vertices in $[n]$. We use the terms nodes and vertices interchangeably. We use bold fonts for matrices and vectors.

We consider several canonical graph matrices associated with $G$. The adjacency matrix $\mathbf{A}$ of $G$ is defined entrywILE by $\mathbf{A}_{uv} = w_{uv}$ for $(u, v) \in E$ and 0 otherwILE. The degree matrix $\mathbf{D}$ is the diagonal matrix with entries $\mathbf{D}_{uu} = \deg(u) = \sum_{v=1}^{n} \mathbf{A}_{uv}$. The graph Laplacian matrix is then given by $\mathbf{L} = \mathbf{D} - \mathbf{A}$. Since $G$ is undirected, the matrices $\mathbf{A}$, $\mathbf{D}$, and $\mathbf{L}$ are all symmetric, and thus admit real eigenvalues with orthonormal eigenvectors. We will generally denote the eigenvalues of any symmetric graph matrices $\mathbf{M}$ (including the graph Laplacian $\mathbf{L}$) by $\lambda_1, \ldots, \lambda_n$ in ascending order ($\lambda_i \leq \lambda_j$ for $i < j$). Following the usual convention of spectral graph theory, for the special case of the adjacency matrix $\mathbf{A}$, we denote its eigenvalues by $\omega_1, \ldots, \omega_n$ in descending order ($\omega_i \geq \omega_j$ for $i < j$).

## 2.2 Spectral Embeddings of Nodes and the Two-Truths Phenomenon

The eigenvectors of graph matrices can be used to construct low-dimensional spectral node embeddings that preserves structural properties of the underlying graph. Consider the eigendecomposition of the Laplacian $\mathbf{L} = \sum_{u=1}^{n} \lambda_u \mathbf{z}_u \mathbf{z}_u^T$. A spectral embedding of dimension $k$ is constructed by selecting the first $k$ eigenvectors of $\mathbf{L}$ that corresponds to the smallest $k$ non-zero eigenvalues. The entries of such eigenvectors are then used as the coordinates in a Euclidean representation. More concretely, the embedding of node $u$ is given by the $u$-th row of $[\mathbf{z}_1, \mathbf{z}_2, \ldots, \mathbf{z}_k] \in \mathbb{R}^{n \times k}$. Adjacency spectral embeddings are constructed analogously, except the eigenvectors corresponding to the *largest* $k$ eigenvalues are selected.

Arguably the most canonical and commonly used graph matrices are the Laplacian matrix $\mathbf{L}$ and adjacency matrix $\mathbf{A}$. Most work in the literature utilizes only one particular type of spectral embedding, usually the Laplacian by default. The effect of matrix choice on downstream tasks such as clustering have only been explicitly studied recently. Notably, the pioneering work of Priebe et al. (2019) identified the "two-truths" phenomenon: spectral embeddings from the normalized Laplacian versus the adjacency matrix yield clustering results that are drastically different yet both valid. Laplacian spectral embeddings capture community structure, which reflects high connectivity within communities and low connectivity between communities. This can be interpreted via the well known relationship between Laplacian eigenvectors and approximate graph cuts (Von Luxburg, 2007). In contrast, adjacency embeddings capture "core-periphery" structure (Priebe et al., 2019). Here, high-degree nodes that are highly connected form a hub, whereas low-degree nodes that are sparsely connected (often only to the core) form the periphery. We pictorially illustrate this phenomenon in Figure 1. While Priebe et al. (2019) used a stochastic blockmodel and asymptotic analysis to explain this phenomenon, in section 3.1 we provide an alternative explanation that does not resort to asymptotics. This phenomenon suggests that the choice of graph matrix for spectral embedding can affect downstream tasks, motivating our study of its impact on node classification with GNNs.

## 2.3 Graph Neural Networks and Spectral Embeddings

Graph neural networks (GNNs) most commonly operate under a message passing scheme, which iteratively applies two general steps: a message-passing step, where nodes aggregate information from their neighbors, and an update step, where nodes update their representations/learned embeddings using that aggregated information. The starting point of such representations is usually taken to be a set of node (and/or edge) features. Stacking $L$ such layers of message passing together yields node representation that encode structural information from neighborhoods up to $L$ hops away. This message passing scheme encompasses a wide variety of popular GNN architectures, including Graph Convolutional Networks (GCNs) (Kipf & Welling, 2017), GraphSAGE (Hamilton et al., 2018), and Graph Isomorphism Networks (GINs) (Xu et al., 2019). The choice of how the messages are passed/aggregated and how the node representations are often what distinguishes GNN models from one another.

Given the importance of node features for GNN learning, in many applications it is often of interest to supplement node features by spectral embeddings (Said et al., 2023; Lim et al., 2022). Since spectral embeddings are eigenvectors of symmetric matrices, they are subject to certain symmetries:

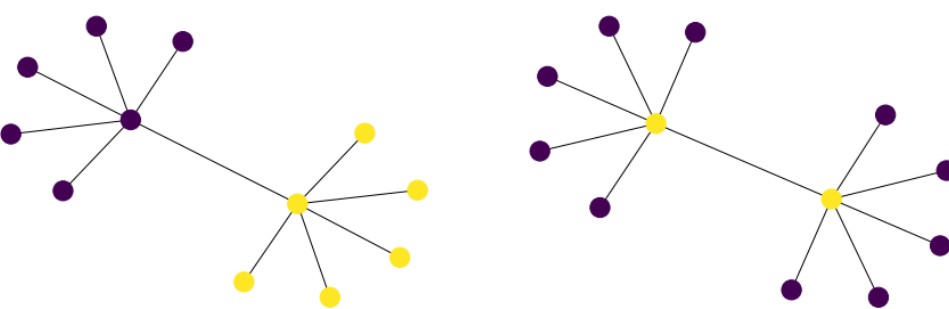

(a) Laplacian embedding ($k = 1$) captures community structure

(b) Adjacency embedding ($k = 1$) captures core-periphery structure

Figure 1: Illustration of the two truths phenomenon on a small graph. Nodes are colored by thresholding the corresponding embedding values.

for any eigenvector $\mathbf{z} \in \mathbb{R}^n$, the negated vector $-\mathbf{z}$ is also an eigenvector associated with the same eigenvalue. Moreover, in cases of eigenvalue multiplicity, the corresponding eigenspace admits infinitely many valid orthonormal bases up to rotations. SignNet and BasisNet (Lim et al., 2022) are general frameworks that allow for the use of spectral embeddings in GNNs while respecting symmetry constraints. SignNet achieves invariance to sign flips by symmetrizing the response of a neural network $\phi$ applied to each eigenvector individually. Given $k$ eigenvectors $\mathbf{z}_1, \ldots, \mathbf{z}_k \in \mathbb{R}^n$, SignNet defines the mapping $f(\mathbf{z}_1, \ldots, \mathbf{z}_k) = \rho\Big(\big[\phi(\mathbf{z}_i) + \phi(-\mathbf{z}_i)\big]_{i=1}^{k}\Big)$, where $\rho$ is an additional learnable aggregation function. By construction, $f$ remains unchanged under any sign flip, thereby ensuring consistency of learned representations across different eigenbasis choices. BasisNet operates analogously for rotations for graphs that have repeated eigenvalues: given a matrix of eigenvectors $\mathbf{Z} = [\mathbf{z}_1, \cdots, \mathbf{z}_k] \in \mathbb{R}^{n \times k}$ as input, BasisNet considers the associated orthogonal projector matrix $\mathbf{Z}\mathbf{Z}^\top$ since it is invariant to rotations. We defer to Lim et al. (2022) for details. Rather than appending spectral embeddings directly as node features, SignNet and/or BasisNet first process the embeddings, and their outputs are then appended as features. While BasisNet is only applicable for eigenvectors with repeated eigenvalues, SignNet should always be used since signflips symmetry affects all eigenvectors.

## 3 INTERPOLATED LAPLACIAN EMBEDDINGS

Motivated by the success of feature augmentation with Laplacian spectral embeddings in GNNs, as well as the observation that different graph matrices can encode different graph structural information, we ask whether spectral embeddings from more general families of graph matrices can further improve performance. To this end, we propose a simple family of graph matrices that subsumes many classical graph matrices as special cases.

### 3.1 A FAMILY OF GRAPH MATRICES

One of the most general families of graph matrices proposed in the literature is the universal adjacency matrices family (Haemers & Omidi, 2011), which takes the form

$$\mathbf{M}(\alpha, \beta, \kappa, \zeta) = \alpha\mathbf{D} + \beta\mathbf{A} + \kappa\mathbf{J} + \zeta\mathbf{I}, \tag{1}$$

where $\mathbf{J}$ is the all-ones matrix, $\mathbf{I}$ is the identity matrix, and $\alpha, \beta, \kappa, \zeta \in \mathbb{R}$. This family has natural mathematical properties and includes many of the most important graph matrices as special cases, such as the Laplacian, the signless Laplacian and the Seidel matrix (Haemers & Omidi, 2011). Despite the richness, the fact that this family depends on four tuning parameters makes it less practical for use in machine learning applications.

Observing that $\mathbf{J}$ and $\mathbf{I}$ are matrices that do not reflect structural information of the underlying graph, we consider the following two-parameter family of graph matrices that we call the interpolated

Laplacians:

$$\mathbf{M}(t, s) \equiv t\mathbf{D} - s\mathbf{A}, \tag{2}$$

where $t, s \in \mathbb{R}$. Here, we drop the $\mathbf{J}$ and $\mathbf{I}$ components from the universal adjacency matrix family, and without loss of generality adopt a subtraction based formulation to mimic the form of the Laplacian. We call spectral embeddings constructed from the eigenvectors of the interpolated Laplacian family of matrices interpolated Laplacian embeddings (ILEs). Below, we interpret the information captured by ILEs.

## 3.2 Interpretation of interpolated Laplacian Embeddings

An important way to interpret eigenvectors is as solutions of certain constrained optimization problems on quadratic forms. Given any symmetric matrix $\mathbf{M} \in \mathbb{R}^{n \times n}$ and nonzero vector $\mathbf{x} \in \mathbb{R}^n$, its Rayleigh quotient is defined as

$$R_{\mathbf{M}}(\mathbf{x}) = \frac{\mathbf{x}^\top \mathbf{M} \mathbf{x}}{\mathbf{x}^\top \mathbf{x}}. \tag{3}$$

Note that without loss of generality, it suffices to consider only unit vectors $\mathbf{x}$, whence $R_{\mathbf{M}}(\mathbf{x})$ is just the quadratic form $\mathbf{x}^\top \mathbf{M} \mathbf{x}$.

Let $\lambda_1 \leq \lambda_2 \leq \cdots \leq \lambda_n$ denote the ordered eigenvalues of $\mathbf{M}$. The Courant–Fischer theorem (Spielman, 2019) states that the extrema of $R_{\mathbf{M}}(\mathbf{x})$ occur at the eigenvectors of $\mathbf{M}$ with the corresponding optimal values being the eigenvalues of $\mathbf{M}$. More concretely, we have the following expressions for the eigenvalues

$$\lambda_n = \max_{\mathbf{x} \neq 0} R_{\mathbf{M}}(\mathbf{x}), \quad \lambda_1 = \min_{\mathbf{x} \neq 0} R_{\mathbf{M}}(\mathbf{x}), \tag{4}$$

$$\lambda_k = \min_{S \in \mathbb{R}^n, \ \dim S = k} \ \max_{\mathbf{x} \in S \setminus \{0\}} R_{\mathbf{M}}(\mathbf{x}) = \max_{S \in \mathbb{R}^n, \ \dim S = n-k+1} \ \min_{\mathbf{x} \in S \setminus \{0\}} R_{\mathbf{M}}(\mathbf{x}), \tag{5}$$

for any $1 \leq k \leq n$. The eigenvectors can be characterized by

$$\mathbf{z}_1 \in \underset{||\mathbf{x}||=1}{\arg\min} \ \mathbf{x}^\top \mathbf{M} \mathbf{x}, \quad \mathbf{z}_k \in \underset{||\mathbf{x}||=1, \ \mathbf{x} \perp \mathbf{z}_1, \cdots, \mathbf{z}_{k-1}}{\arg\min} \ \mathbf{x}^\top \mathbf{M} \mathbf{x}, \tag{6}$$

$$\mathbf{z}_k \in \underset{||\mathbf{x}||=1, \ \mathbf{x} \perp \mathbf{z}_{k+1}, \cdots, \mathbf{z}_n}{\arg\max} \ \mathbf{x}^\top \mathbf{M} \mathbf{x}, \tag{7}$$

for any $2 \leq k \leq n$.

Thus, we can interpret the first $k$ eigenvectors of $\mathbf{M}$ as a set of vectors that collectively minimizes Rayleigh quotient. This optimization perspective provides a way for us to interpret the resulting embeddings.

**Interpreting $\mathbf{M}(t, s)$ via quadratic forms** Consider the matrix family $\mathbf{M}(t, s) = t\mathbf{D} - s\mathbf{A}$, $t, s \in \mathbb{R}$. For any unit vector $\mathbf{x} \in \mathbb{R}^n$, its quadratic form is given by

$$\mathbf{x}^\top \mathbf{M}(t, s)\mathbf{x} = t\mathbf{x}^\top \mathbf{D} x - s\mathbf{x}^\top \mathbf{A}\mathbf{x}. \tag{8}$$

Expanding each term:

$$\mathbf{x}^\top \mathbf{D} \mathbf{x} = \sum_{i=1}^n \deg(i)\mathbf{x}_i^2 = \sum_{(i,j) \in E} w_{ij} \left(\mathbf{x}_i^2 + \mathbf{x}_j^2\right), \tag{9}$$

$$\mathbf{x}^\top \mathbf{A} \mathbf{x} = \sum_{i,j} A_{ij}\mathbf{x}_i\mathbf{x}_j = 2 \sum_{(i,j) \in E} w_{ij} \, \mathbf{x}_i\mathbf{x}_j. \tag{10}$$

This gives

$$\mathbf{x}^\top \mathbf{M}(t, s)\mathbf{x} = \sum_{(i,j) \in E} w_{ij} \left[ t(\mathbf{x}_i^2 + \mathbf{x}_j^2) - 2s\mathbf{x}_i\mathbf{x}_j \right] = \sum_{(i,j) \in E} w_{ij} \left[ t(\mathbf{x}_i - \mathbf{x}_j)^2 - 2(s - t)\mathbf{x}_i\mathbf{x}_j \right].$$
$$\tag{11}$$

This makes apparent that the objective combines the terms $\sum_{(i,j)\in E} w_{ij}(\mathbf{x}_i - \mathbf{x}_j)^2$ and $2\sum_{(i,j)\in E} w_{ij}\mathbf{x}_i\mathbf{x}_j$, which enables various tradeoffs.

It is well known that the term $\sum_{(i,j)\in E} w_{ij}(\mathbf{x}_i - \mathbf{x}_j)^2$ is equal to the Laplacian quadratic form $\mathbf{x}^\top \mathbf{L}\mathbf{x}$ (Spielman, 2019; Von Luxburg, 2007). This objective penalizes variation of node embeddings across heavily weighted edges, encouraging strongly connected nodes to take similar values and thereby capturing community structure in the graph.

On the other hand, $2\sum_{(i,j)\in E} w_{ij}\mathbf{x}_i\mathbf{x}_j$ is equal to the adjacency quadratic form $\mathbf{x}^\top \mathbf{A}\mathbf{x}$. Maximizing this objective (equivalent to minimizing its negation in equation 11) corresponds to assigning larger embedding values for nodes that have a high degree of connectivity (large degree), thus capturing core-periphery structures. This provides an alternative explanation of adjacency embeddings without resorting to the asymptotic arguments and modeling assumptions used in Priebe et al. (2019). Taken together, these characterizations show that varying $t$ and $s$ yields embeddings that balance core–periphery and community structure differently. If node labels in a classification task depend on some combination of these structures, then spectral embeddings from a suitably chosen $\mathbf{M}(s,t)$ may offer an advantage.

### 3.3 RICHNESS OF REPRESENTATION

In defining the matrix family $\mathbf{M}(t,s)$, we have dropped two components corresponding to $\mathbf{I}$ and $\mathbf{J}$ in the universal adjacency matrix family $\mathbf{M}(\alpha,\beta,\kappa,\zeta)$ for a more parsimonious representation that is conducive to ML practice. Below, we show that dropping the $\mathbf{I}$ component does not affect the richess of our spectral embeddings. This is exemplified by the following lemma:

**Lemma 3.1.** *Let $\mathbf{M} \in \mathbb{R}^{n\times n}$ be a symmetric matrix with no repeated eigenvalues, and consider its perturbation $\mathbf{M} + \zeta\mathbf{I}$ for some scalar $\zeta \in \mathbb{R}$. Then the eigenvectors of $\mathbf{M}$ and $\mathbf{M} + \zeta\mathbf{I}$ coincide, and the eigenvalues of $\mathbf{M} + \zeta\mathbf{I}$ are shifted by $\zeta$ relative to their counterparts in $\mathbf{M}$.*

*Proof.* Let $(\lambda, \mathbf{x})$ be an eigenpair of $\mathbf{M}$, i.e. $\mathbf{M}\mathbf{x} = \lambda\mathbf{x}$ where, $\mathbf{x} \neq 0$. Then $(\mathbf{M} + \zeta\mathbf{I})\mathbf{x} = \mathbf{M}\mathbf{x} + \zeta\mathbf{I}\mathbf{x} = (\lambda + \zeta)\mathbf{x}$. Thus $\mathbf{x}$ is also an eigenvector of $\mathbf{M} + \zeta\mathbf{I}$ with eigenvalue $\lambda + \zeta$. $\square$

Note that the same statement applies when $\mathbf{M}$ has repeated eigenvalues, except it would be the eigenspaces rather than then eigenvectors that are identical.

The above implies that not only are the eigenvectors of $\mathbf{M}$ unchanged when it is perturbed by $\zeta\mathbf{I}$, the ordering of the eigenvalues are unchanged either. In our context of selecting the first $k$ eigenvectors for spectral embeddings, the above lemma shows that $\mathbf{M}(t,s)$ yields the same expressivity of spectral embeddings as $\mathbf{M}(t,s) + \zeta\mathbf{I}$.

While it is immediate that certain natural choices of $(t,s)$ reduce to well-known matrices (e.g., $t = 1, s = 1$ for the Laplacian, $t = 0, s = -1$ for the adjacency, and $t = 1, s = -1$ for the sign-less Laplacian), Lemma 3.1 allows us to show the less obvious fact that embeddings from $\mathbf{M}(t,s)$ subsume the embeddings generated by the deformed Laplacian graph matrix family.

Recall the deformed Laplacian matrices (Grindrod et al., 2018)

$$\mathbf{L}_{\text{deformed}}(q) = \mathbf{I} - q\mathbf{A} + q^2(\mathbf{D} - \mathbf{I}), \quad q \in \mathbb{R}, \tag{12}$$

Note that in the original formulation $q \in \mathbb{C}$ is allowed for full mathematical generality, here we only consider real $q$ for our ML context. This family of matrices is important in graph theory, as it is known to capture meaningful centrality measures via non-backtracking random walks (Grindrod et al., 2018).

Expanding, we obtain

$$\mathbf{L}_{\text{deformed}}(q) = (1 - q^2)\mathbf{I} + q^2\mathbf{D} - q\mathbf{A}.$$

Thus, setting $t = q^2$ and $s = q$, and applying Lemma 3.1, we see that the eigenvectors and thus spectral embeddings of $\mathbf{M}(t,s)$ contains those generated by $\mathbf{L}_{\text{deformed}}(q)$.

The above results demonstrate the richness of the interpolated Laplacian family $\mathbf{M}(t,s)$. We next consider testing this family in empirical experiments.

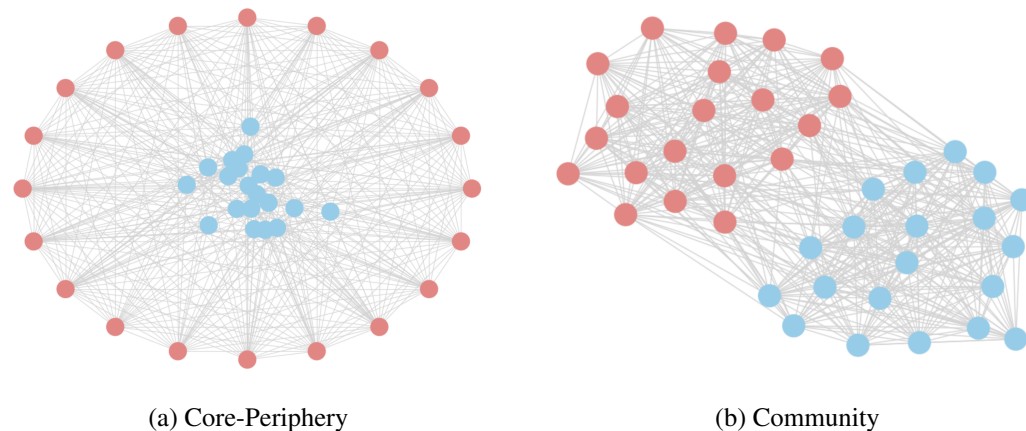

(a) Core-Periphery  (b) Community

Figure 2: Illustration of two realizations of SBM graphs that exhibit (a) core-periphery and (b) community structures

# 4 EXPERIMENTS AND SIMULATIONS

We empirically test the hypothesis that alternative spectral embeddings can potentially lead to downstream performance gains compared to the Laplacian case.

## 4.1 EXPERIMENTAL SETUP

We evaluate the effectiveness of ILE in the context of node classification, where ILE is used to augment node features in GNNs. The resulting models are then assessed based on their classification accuracy. Our experiments cover a range of datasets and models.

We consider three settings.

1. We first consider a synthetic scenario with Stochastic Block Models (SBM) consisting of two blocks of equal size. Node labels correspond to block membership. Following Priebe et al. (2019), we choose connectivity parameters that lead to two graphs exhibiting core–periphery and community structures respectively. See Figure 2 for illustration.

2. We consider network datasets that do not come with node features. This include the Karate Club (Rozemberczki et al., 2020), Twitter Congress (Fink et al., 2023), Facebook Ego (Leskovec & Mcauley, 2012), and Political Blogs (Polblogs) (Adamic & Glance, 2005) datasets. We use ILEs as node features on these datasets.

3. We consider network datasets that come with node features, which include the Cornell (Pei et al., 2020), Texas (Pei et al., 2020), and Wisconsin (Pei et al., 2020) datasets. We corrupt the original node features with various levels of noise, and append ILEs as extra node features to observe performance.

We consider three representative models: GCN Kipf & Welling (2017), GIN Xu et al. (2019), and GraphSAGE Hamilton et al. (2018). We also use a 5-layer MLP as a baseline. Detailed descriptions of all datasets are provided in Appendix B.

We adopt a train-test split using a 70/30 ratio. To account for stochasticity, each experiment is repeated 5 times, and we report the mean and standard deviation of the resulting classification accuracies.

We try ILEs with $s$ taking values in the range $[-1, -0.5, 0, 0.5, 1]$ and $t$ taking values in the range $[-1, -0.5, 0.5, 1]$. The case $s = 1, t = 1$ corresponds to the Laplacian case. We omit the case of $t = 0$ since it corresponds to various scalings of the adjacency matrix.

## 4.2 SBM Simulations

As shown in Table 1, the best-performing embeddings across the datasets are members of ILEs that do not correspond to Laplacian embeddings. Observe that the GCN model has the best performance, whereas the MLP model, which did not leverage graph convolutional operations, performed roughly at 50 percent accuracy. Since the Core-Periphery case is largely determined by the degree representation, we would expect that the embeddings that are most informative about high degree nodes ($t = 1$) or low degree nodes ($t = -1$)) to enjoy good performance. Indeed, for GCN, GIN and GraphSAGE, the best performance came from the $t = -1$ and $t = 1$ columns.

In the community structure case, we would expect that embeddings that are close to the Laplacian or Signless Laplacian embeddings would capture the community structure much better than adjacency based embeddings. Indeed, across the models the ILE embeddings performed better than the adjacency embeddings. Note that GCN is a special case since its convolutional operation includes a normalized Laplacian component, so it is unsurprising that it is able to detect community patterns better than other models.

Table 1: Testing accuracy (%) of different models augmented with spectral embedding variants under SBM graphs that exhibit core-periphery and community structure.

| Model | Variant | SBM Core-Periphery | | | | SBM Community | | | |
|---|---|---|---|---|---|---|---|---|---|
| | | $t = -1$ | $t = -0.5$ | $t = 0.5$ | $t = 1$ | $t = -1$ | $t = -0.5$ | $t = 0.5$ | $t = 1$ |
| GCN | None | 45.33(4.64) | | | | 48.00(1.94) | | | |
| | Adjacency | 94.33(3.43) | | | | 99.67(0.67) | | | |
| | $s = -1$ | 94.33(4.03) | 94.67(1.94) | 93.67(1.94) | 92.00(4.00) | 96.67(4.35) | 99.00(0.82) | 96.00(6.38) | 99.67(0.67) |
| | $s = -0.5$ | 94.33(4.16) | 89.00(4.67) | 95.00(2.36) | 92.00(1.25) | 99.33(1.33) | 98.67(1.25) | 99.67(0.67) | 94.67(9.85) |
| | $s = 0$ | **97.00(1.94)** | 88.00(4.00) | 93.33(2.36) | 91.67(3.50) | 99.33(0.82) | 98.33(1.49) | 98.67(1.25) | **100.00(0.00)** |
| | $s = 0.5$ | 96.00(3.27) | 91.67(3.80) | 91.67(3.50) | 92.33(4.90) | 96.33(6.53) | 97.00(4.40) | **100.00(0.00)** | 99.33(0.82) |
| | $s = 1$ | 93.33(1.83) | 92.67(4.29) | 91.67(2.98) | 95.33(2.87) | 72.67(25.66) | 98.00(4.00) | 96.00(3.74) | 98.33(0.00) |
| MLP | None | 43.67(3.23) | | | | 45.33(2.87) | | | |
| | Adjacency | 45.67(3.43) | | | | 45.67(3.74) | | | |
| | $s = -1$ | 52.33(3.27) | **57.33(3.43)** | 48.33(4.35) | 50.33(7.18) | 52.33(6.11) | 47.33(6.80) | 48.00(5.52) | **55.33(4.14)** |
| | $s = -0.5$ | 50.00(6.83) | 54.67(3.23) | 48.67(9.85) | 47.67(8.79) | 47.00(5.62) | 49.33(7.57) | 50.00(3.33) | 45.67(4.55) |
| | $s = 0$ | 45.67(5.12) | 49.33(5.44) | 46.33(5.31) | 51.33(3.56) | 51.67(3.33) | 50.00(7.67) | 47.67(4.78) | 47.00(6.00) |
| | $s = 0.5$ | 50.00(7.89) | 48.67(5.42) | 47.67(7.20) | 47.00(6.53) | 51.00(8.00) | 54.00(7.93) | 50.67(6.55) | 51.67(6.41) |
| | $s = 1$ | 52.33(4.90) | 51.33(3.23) | 51.67(6.50) | 49.33(4.67) | 50.00(5.96) | 46.67(4.22) | 51.33(5.81) | 51.00(4.67) |
| GIN | None | 50.33(4.00) | | | | 43.00(6.18) | | | |
| | Adjacency | 74.67(22.98) | | | | 84.67(15.43) | | | |
| | $s = -1$ | 77.67(16.62) | 74.00(23.49) | 68.00(15.54) | 83.00(17.04) | 87.67(10.62) | 91.00(6.63) | 87.33(9.64) | 87.67(16.62) |
| | $s = -0.5$ | 87.00(6.09) | 82.00(13.43) | 73.67(23.46) | **91.67(3.16)** | 91.00(6.72) | 83.00(17.90) | 89.33(6.02) | 83.67(19.45) |
| | $s = 0$ | 79.00(14.67) | 77.67(18.15) | 80.00(17.09) | 88.00(10.67) | **96.67(3.80)** | 79.67(17.99) | 91.33(4.88) | 75.67(22.45) |
| | $s = 0.5$ | 81.00(16.28) | 88.33(13.00) | 81.00(19.91) | 78.67(19.04) | 95.33(2.87) | 93.67(4.14) | 92.00(8.72) | 96.00(1.33) |
| | $s = 1$ | 67.33(17.11) | 81.33(10.97) | 76.67(18.41) | 54.67(20.61) | 94.67(6.78) | 96.00(2.26) | 92.33(5.01) | 94.00(3.59) |
| GraphSAGE | None | 48.33(4.47) | | | | 48.33(5.48) | | | |
| | Adjacency | 60.00(8.88) | | | | 56.00(10.62) | | | |
| | $s = -1$ | 57.00(16.78) | 77.00(19.76) | 66.67(18.35) | 70.00(13.82) | 93.33(6.58) | 90.67(5.01) | 86.67(6.99) | 86.00(8.67) |
| | $s = -0.5$ | 68.00(18.18) | 73.00(4.76) | 79.33(11.58) | 75.00(20.52) | 85.33(7.99) | 84.33(12.85) | 84.33(12.36) | 87.00(7.99) |
| | $s = 0$ | 72.67(12.00) | 75.00(13.98) | 82.67(12.54) | 70.00(17.48) | 88.33(10.70) | 89.33(5.12) | 83.33(4.71) | 84.33(10.09) |
| | $s = 0.5$ | **88.67(8.06)** | 73.00(14.73) | 71.00(14.55) | 79.67(13.60) | 81.00(16.38) | 85.00(9.94) | 91.67(5.48) | **93.67(4.40)** |
| | $s = 1$ | 66.67(8.94) | 71.33(10.82) | 69.33(17.34) | 67.67(9.35) | 83.67(8.39) | 91.67(6.75) | 81.00(8.07) | 80.00(16.30) |

## 4.3 Datasets without Features

In our experiments with the datasets that did not come with node features, we again consistently see the pattern where spectral feature augmentation with ILEs led to better results than the Laplacian embeddings ($s = 1, t = 1$). Results are shown in Table 2 in the appendix.

## 4.4 Datasets with Node Features Corrupted by noise

We also considered three datasets that come with node features (Texas, Cornell and Wisconsin). Here we corrupt the original node features with Gaussian noise of different magnitude. Results for the datasets Texas, Cornell and Wisconsin are shown in Tables 5, 3 and 4, respectively, in the appendix. We observe a pattern where the larger the noise perturbation, generally the worse the performance. We also observe that at low noise corruption levels, the ILE families that perform well in general (e.g. the $t = 0.5$ column in the Wisconsin dataset under 0 percent and 10 percent corruption in Table 4) are generally stable. At high noise corruption level, the numerous noisy node features appear to overpower the effect of the spectral embeddings.

## 4.5 SELECTING HYPERPARAMETERS

A main goal of this paper is to show that more general spectral embeddings can outperform Laplacian embeddings. While this observation holds true in our experiments which look at options in an exhaustive fashion, in practice users might have to select certain tuning parameters, including the dimension of the spectral embeddings $k$, as well as the tuning parameters $t, s$ for the interpolated Laplacian family.

For selecting the spectral dimension $k$ for Laplacian-based families, the multi-way Cheeger's inequality Lee et al. (2014) suggests that $k$ should be selected as the number of communities in the underlying graph. If no such prior information is available, then a conservative $k$ (an upper bound on the number of communities) should be chosen. For general graph matrices, one can adopt the commonly used strategy of inspecting Scree plots and determining $k$ according to the location of the elbow (Cattell, 1966).

As for the selection of $s, t$, if ones has prior information on the graph's underlying structure (e.g. community versus core-periphery) and their relationship to the node labels, one can selectively try appropriate ranges of $s, t$ according to the interpretation provided in Section 3. A slightly more computationally intensive approach would be to compute some correlation measure between the spectral embeddings and the node labels in a training or validation set. Yet another more computationally intensive approach is to perform cross-validation to select the $s, t$ that leads to the highest validation accuracy.

## 4.6 COMPUTATIONAL TIME

The computational complexity of a naïve spectral decomposition of a $n \times n$ matrix is on the order of $O(n^3)$. However, fast numerical and/or iterative schemes (e.g. Lanczos algorithm, Power method, Preconditioned methods etc) are often available and can be highly effective in practice for the computational of only the top $k$ eigenvectors (Press, 2007). Some of these iterative methods enjoy linear converge rates. Moreover, fast numerical methods are often available for sparse graphs or for the approximate computational of eigenvectors, which may suffice for the application at hand.

## 5 DISCUSSION

This work broadens the scope of spectral augmentation in graph neural networks by moving beyond the Laplacian to a unified family of graph matrices that we term interpolated Laplacian Embeddings (ILEs). Our theoretical analysis provides an interpretable characterization of the structural properties that different parameterizations of ILEs capture, ranging from community to core–periphery structure. Empirically, we demonstrated that the choice of graph matrix can substantially influence downstream node classification performance, particularly in the limited- or missing-feature regime. Natural directions of future research include generalization of such approaches to directed graphs, or to graphs with edge features, and to graphs with dynamic, temporal, or heterophilic structures. The theoretical and empirical aspects of non-Laplacian operators in graph machine learning in general is understudied compared to the canonical Laplacian case. We hope that this work motivates more systematic study of spectral embeddings beyond the Laplacian, and inspires new methods that more effectively leverage different types of graph structures for representation learning.

## ETHICS STATEMENT

We adhere to the ICLR Code of Ethics and report no ethical concerns in producing this work.

## REPRODUCIBILITY STATEMENT

We attach all code that generated the experimental results as supplementary material. The code is commented for interpretability and seeded for reproducibility. All datasets that we use are publicly available. We describe experimental and dataset details in the appendix and in section 4 of the main paper. For theoretical results, all proofs are provided in the paper and relevant references cited.

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

## A LLM STATEMENT

Language models are used to check for grammatical mistakes. Language models were also used to select appropriate wordings and improve sentence flow.

## B  DATASET DETAILS

We first evaluate our family of spectral embeddings on synthetic networks generated from SBMs. The SBM is a fundamental generative model for random graphs with community or group structure. In its general form, the model partitions nodes into latent blocks and specifies the probability of an edge between any two nodes as a function of their block memberships. By varying the block structure and edge probabilities, the SBM is able to capture a wide range of network patterns. In our simulations, we generate networks with 1000 nodes. To study a core–periphery structure, we divide the nodes evenly into a 500-node core and a 500-node periphery. Edges between core nodes are drawn with high probability (90%), edges between core and periphery nodes with intermediate probability (50%), and edges between periphery nodes with low probability (10%). To study a community structure, we again partition the 1000 nodes into two equal groups of 500. In this case, the within-group edge probability is very high (99%), while the between-group probability is comparatively low (30%). Figure 2 illustrates example realizations of the core–periphery and community SBM graphs using 40 nodes.

We next evaluate ILE on real-world networks that do not contain node features. In these settings, we demonstrate that augmenting the nodes with ILE substantially improves classification accuracy. The datasets in this group are Zachary's Karate Club (Rozemberczki et al., 2020), a social network of 34 nodes and 78 edges partitioned into four classes; Twitter Congress (Fink et al., 2023), a network of 475 nodes and 13,289 edges representing interactions among members of the 117th United States Congress; Facebook Ego (Leskovec & Mcauley, 2012), a social network of 4,039 nodes and 88,234 edges capturing ego-centric friendship circles on Facebook; and Political Blogs (Polblogs) (Adamic & Glance, 2005), a network of 1,490 nodes and 19,025 edges, with binary labels indicating the political leaning of online blogs. The Twitter Congress and Facebook Ego datasets lack ground-truth node labels. To enable evaluation, we construct artificial labels based on node degree: nodes in the top 20% by degree are assigned label 1, and all others are assigned label 0.

Finally, we evaluate the ILE family on real-world datasets with node attributes. In this setting, we show that augmenting node features with ILE improves classification accuracy both under standard conditions and in scenarios where the original features are corrupted. The datasets in this group are Cornell (Pei et al., 2020), a web network with 183 nodes, 298 edges, 1,703 features, and five classes, where nodes correspond to university web pages and edges represent hyperlinks; Texas Pei et al. (2020), a structurally similar university web network with 183 nodes, 325 edges, 1,703 features, and five classes; and Wisconsin (Pei et al., 2020), another university web network with 251 nodes, 515 edges, 1,703 features, and five classes.

## C  FURTHER EXPERIMENTAL RESULTS

Table 2: Testing accuracy (%) of different models augmented with spectral embedding variants under various datasets that did not come with node features.

| Model | Variant | Karate Club | | | | Twitter Congress | | | | Facebook Ego | | | | Polblogs | | | |
|---|---|---|---|---|---|---|---|---|---|---|---|---|---|---|---|---|---|
| | | $t=-1$ | $t=-0.5$ | $t=0.5$ | $t=1$ | $t=-1$ | $t=-0.5$ | $t=0.5$ | $t=1$ | $t=-1$ | $t=-0.5$ | $t=0.5$ | $t=1$ | $t=-1$ | $t=-0.5$ | $t=0.5$ | $t=1$ |
| GCN | None | 29.09(8.91) | | | | 79.58(2.23) | | | | 79.55(0.77) | | | | 51.41(0.78) | | | |
| | Adjacency | 92.73(8.91) | | | | 84.76(2.36) | | | | 87.34(0.84) | | | | 83.45(4.46) | | | |
| | $s=-1$ | 92.73(6.80) | 81.82(9.96) | 76.36(7.27) | | 85.17(2.14) | 86.71(2.12) | 86.15(2.70) | 86.57(1.68) | 87.01(0.84) | 86.91(1.29) | 86.20(0.74) | 87.26(0.74) | 75.53(5.31) | 79.91(4.14) | 70.02(8.29) | 79.06(5.08) |
| | $s=-0.5$ | 83.64(17.63) | 78.18(7.27) | 85.45(12.33) | 83.64(6.80) | 85.03(1.05) | 81.96(2.27) | 86.43(2.45) | 87.41(1.71) | 87.38(0.88) | 86.70(0.51) | 87.06(0.09) | 79.11(3.78) | 79.02(3.11) | 79.73(7.91) | 70.96(7.98) |
| | $s=0$ | 78.18(7.27) | 78.18(10.91) | 80.00(6.80) | 72.73(16.26) | 85.31(2.12) | 86.43(2.41) | 84.62(2.42) | 84.76(1.36) | 86.25(0.43) | 87.15(0.68) | 86.50(0.97) | 86.53(0.53) | 74.18(6.69) | 82.19(2.55) | 73.74(9.78) | 79.82(1.91) |
| | $s=0.5$ | 85.45(7.27) | 87.27(4.45) | 69.09(30.21) | 89.09(10.60) | 86.29(1.05) | 84.34(1.80) | 85.31(2.69) | 85.31(2.50) | 86.52(0.21) | 86.91(0.59) | 87.29(0.67) | 87.59(0.72) | 78.75(5.32) | 84.52(1.11) | 75.26(6.54) | 77.05(7.76) |
| | $s=1$ | 83.64(12.06) | 92.73(6.80) | 89.09(6.80) | 89.09(6.80) | 85.45(2.23) | 86.57(2.56) | 85.31(1.53) | 87.83(2.75) | 86.44(0.42) | 86.98(0.86) | 86.98(0.34) | 87.49(0.36) | 77.27(5.99) | 75.26(1.10) | 76.02(8.88) | 76.38(4.04) |
| MLP | None | 32.73(12.33) | | | | 77.90(1.80) | | | | 79.42(0.73) | | | | 50.07(2.55) | | | |
| | Adjacency | 80.00(21.82) | | | | 83.22(3.62) | | | | 80.17(1.38) | | | | 49.57(2.13) | | | |
| | $s=-1$ | 61.82(18.54) | 63.64(9.96) | 70.91(10.60) | 63.64(9.96) | 73.99(2.09) | 74.55(4.57) | 79.58(2.56) | 78.60(1.86) | 79.92(0.95) | 79.69(0.74) | 79.31(0.85) | 79.62(0.62) | 56.02(5.00) | 54.77(1.63) | 50.78(3.22) | 55.97(2.04) |
| | $s=-0.5$ | 58.18(18.72) | 69.09(12.33) | 52.73(6.80) | 29.09(10.60) | 78.88(1.03) | 76.92(2.34) | 76.08(4.73) | 81.54(2.01) | 79.88(0.80) | 80.10(0.82) | 79.85(0.82) | 79.70(1.29) | 53.29(2.67) | 55.17(3.29) | 57.94(3.49) | 55.75(3.95) |
| | $s=0$ | 32.73(7.27) | 25.45(6.80) | 43.64(14.55) | 29.09(19.41) | 79.30(3.79) | 78.88(2.85) | 78.60(2.10) | 81.54(2.53) | 80.02(0.56) | 79.55(0.55) | 79.90(0.83) | 80.13(0.93) | 50.20(0.94) | 49.26(2.48) | 48.81(2.40) | 47.61(1.82) |
| | $s=0.5$ | 41.82(9.27) | 54.55(5.75) | 67.27(10.91) | 56.36(21.82) | 78.60(2.82) | 79.16(5.87) | 80.70(1.80) | 78.18(1.12) | 80.31(0.96) | 80.31(0.71) | 80.07(0.40) | 79.87(1.04) | 56.73(1.29) | 57.72(1.87) | 57.39(2.68) | 55.93(0.94) |
| | $s=1$ | 49.09(14.77) | 65.45(10.60) | 61.82(6.80) | 67.27(12.33) | 77.48(4.27) | 74.13(4.00) | 78.46(3.81) | 75.66(2.63) | 79.83(0.71) | 79.55(0.91) | 80.25(0.73) | 79.70(0.51) | 56.60(0.97) | 51.86(3.17) | 57.18(3.43) | 51.68(0.88) |
| GIN | None | 27.27(11.50) | | | | 71.75(13.54) | | | | 71.25(17.65) | | | | 55.84(3.85) | | | |
| | Adjacency | 49.09(9.27) | | | | 85.03(2.60) | | | | 88.07(1.32) | | | | 69.62(11.15) | | | |
| | $s=-1$ | 43.64(31.18) | 67.27(29.65) | 67.27(15.85) | 52.73(36.09) | 86.57(4.04) | 84.90(1.44) | 83.92(2.80) | 84.90(2.24) | 88.73(0.85) | 89.74(0.66) | 89.97(0.49) | 88.91(0.43) | 81.39(4.72) | 74.41(9.44) | 81.12(1.25) | 81.30(2.60) |
| | $s=-0.5$ | 70.91(13.36) | 54.55(22.27) | 63.64(18.18) | 69.09(14.77) | 82.66(3.52) | 85.87(2.70) | 82.66(2.78) | 83.64(2.06) | 88.80(1.29) | 89.19(0.64) | 89.08(0.25) | 88.40(0.48) | 80.98(2.13) | 76.02(8.11) | 80.09(5.21) | 82.51(1.86) |
| | $s=0$ | 52.73(14.55) | 69.09(12.33) | 40.00(25.45) | 65.45(14.55) | 84.34(1.14) | 84.20(2.75) | 82.38(2.59) | 85.03(2.24) | 87.97(1.62) | 88.23(1.11) | 89.09(0.76) | 88.17(1.02) | 79.91(4.54) | 73.78(6.47) | 81.12(1.06) | 80.81(0.67) |
| | $s=0.5$ | 70.91(10.60) | 60.00(24.80) | 72.73(29.88) | 58.18(23.43) | 83.36(5.58) | 83.92(2.21) | 85.31(1.77) | 85.03(1.69) | 89.26(0.67) | 89.27(0.52) | 88.73(0.56) | 88.33(0.61) | 81.25(1.35) | 82.24(2.06) | 82.86(0.97) | 79.42(4.08) |
| | $s=1$ | 49.09(29.65) | 69.09(7.27) | 58.18(29.09) | 60.00(25.45) | 86.15(1.95) | 83.78(0.93) | 85.17(3.20) | 84.90(1.05) | 88.71(1.65) | 88.42(0.96) | 88.99(0.91) | 88.83(0.52) | 76.64(7.55) | 75.84(7.50) | 73.83(7.66) | 81.92(2.25) |
| GraphSAGE | None | 18.18(5.75) | | | | 68.39(23.71) | | | | 79.98(0.91) | | | | 49.98(5.23) | | | |
| | Adjacency | 72.73(22.27) | | | | 59.30(29.94) | | | | 80.48(1.10) | | | | 84.56(2.29) | | | |
| | $s=-1$ | 78.18(17.81) | 80.00(10.60) | 89.09(6.80) | 78.18(18.72) | 80.70(3.11) | 75.38(6.37) | 77.90(3.58) | 82.80(2.68) | 71.37(13.99) | 80.74(0.66) | 76.68(7.76) | 70.33(14.01) | 62.28(5.30) | 62.28(2.88) | 60.40(1.56) | 61.34(4.79) |
| | $s=-0.5$ | 80.00(12.06) | 80.00(10.60) | 83.64(6.80) | 70.91(21.82) | 78.88(2.63) | 81.82(1.47) | 79.72(1.40) | 80.42(2.12) | 79.32(0.47) | 78.81(2.47) | 77.84(3.67) | 74.70(11.67) | 60.54(2.66) | 57.94(0.91) | 62.24(1.67) | 59.37(3.03) |
| | $s=0$ | 65.45(10.60) | 70.91(22.56) | 81.82(28.17) | 76.36(4.45) | 79.72(2.38) | 80.56(4.13) | 81.68(2.59) | 80.28(3.35) | 76.07(9.23) | 79.67(0.85) | 74.42(10.38) | 73.47(9.80) | 54.59(6.69) | 58.21(8.02) | 64.92(4.47) | 53.87(5.21) |
| | $s=0.5$ | 78.18(12.33) | 78.18(12.33) | 81.82(8.13) | 70.91(13.36) | 79.30(3.02) | 79.86(1.62) | 82.80(2.78) | 78.46(3.89) | 75.89(8.50) | 76.47(8.32) | 73.83(11.89) | 74.03(13.10) | 59.69(6.22) | 60.81(7.65) | 59.28(2.73) | 55.35(4.74) |
| | $s=1$ | 76.36(4.45) | 96.36(4.45) | 81.82(23.71) | 80.00(15.64) | 80.42(2.21) | 76.92(2.54) | 77.90(3.55) | 80.42(3.22) | 79.97(0.71) | 79.55(0.26) | 76.53(4.77) | 73.63(11.48) | 64.03(5.27) | 59.73(3.86) | 60.45(6.65) | 60.54(6.54) |

Table 3: Testing accuracy (%) of different models augmented with spectral embedding variants under various feature corruption ratios on the Cornell dataset.

Table 4: Testing accuracy (%) of different models augmented with spectral embedding variants under various feature corruption ratios on the Wisconsin dataset.

Table 5: Testing accuracy (%) of different models augmented with spectral embedding variants under various feature corruption ratios on the Texas dataset.

