# OpenReview forum: "Beyond the Laplacian: Interpolated Spectral Augmentation for Graph Neural Networks"
_ICLR.cc/2026/Conference — Submitted to ICLR 2026_

### Official Review · Reviewer_A9i6 · 2025-10-16

**Soundness:** 2
**Presentation:** 1
**Contribution:** 2
**Rating:** 2
**Confidence:** 3

**Summary:**

This paper introduces Interpolated Laplacian Embeddings (ILE), a two-parameter family of spectral operators M(t,s)=tD−sA, which generalizes the standard Laplacian and adjacency spectra. The authors argue that different t,s combinations capture community versus core–periphery structures and can thus serve as adaptive spectral features for node classification. Experiments on synthetic and small real graphs show that ILE can outperform conventional Laplacian embeddings.

**Strengths:**

The paper presents a framework which is lightweight, broadly applicable to existing GNNs, and empirically demonstrates some improvement.

**Weaknesses:**

- Regarding the motivation of this paper, while the paper highlights the different structural preferences of Laplacian vs. adjacency spectra, it lacks structure–label linkage evidence, such as datasets exhibit community-like vs. core–periphery-like label structures?
- Multiple typos such as wILE, otherwILE, arILE (unclear meaning), richess → richness, and computational of eigenvectors → computation of eigenvectors.
- Experiments:
    - Need strong degree/centrality baseline**s** to ensure gains are not from trivial structure cues; better to include some recent spectral augmentation baselines.
    - No statistical significance tests.
- Some recent spectral augmentation for graph data works [1–6] are not discussed or compared, which are highly-related to this work.

**Reference**:

[1] Revisiting Graph Contrastive Learning from the Perspective of Graph Spectrum. NeurIPS. 2022.

[2] Spectral Feature Augmentation for Graph Contrastive Learning and Beyond. AAAI. 2023.

[3] Spectral Augmentations for Graph Contrastive Learning. AISTATS. 2023.

[4] Through the Dual-Prism: A Spectral Perspective on Graph Data Augmentation for Graph Classifications. AAAI. 2025.

[5] Rethinking Spectral Augmentation for Contrast-based Graph Self-Supervised Learning. 2024.

[6] Spectrum Guided Topology Augmentation for Graph Contrastive Learning. 2022.

**Questions:**

See Weaknesses.

---

### Official Review · Reviewer_NDbB · 2025-10-27

**Soundness:** 2
**Presentation:** 3
**Contribution:** 1
**Rating:** 2
**Confidence:** 4

**Summary:**

This paper introduces a graph positional embedding/encoding (PE) that can be used as additional features to be used by a GNN model. The encoding is a spectral encoding from a matrix interpolating between the Laplacian and the adjacency matrix, where the interpolation coefficients can be chosen by the used depending on the data at hand. The validity of the new positional encoding is showcased on some synthetic data as well as on some established graph datasets.

**Strengths:**

- The idea behind the paper is simple yet elegant and intuitive. The paper does a good job at describing the motivation behind the method and and its intuition.
- The paper is quite well-written.

**Weaknesses:**

The experimental validation, which is crucial for validating the effectiveness of a method such as the one presented here, is extremely weak.
- The paper fails to compare to any baseline method.
    - The simplest baseline would be to concatenate the Laplacian PE with the one obtained from the adjacency matrix, as this would give both community information and Core-Periphery information, while being simpler and requiring no hyperparameters.
    - There is a plethora of PE methods that are not included here. Examples are: ElasticPE, RWSE, SignNet, HKdiagSE. How does the proposed method compare to these methods?
    - Hows does the proposed method compare to learned embeddings? For example, [1] learns the importance of core-periphery information and selects accordingly which eigenvectors to consider.
- The paper only uses few datasets. In particular, in the main paper, only results on synthetic data are reported. The results on real world data should be included in the main text, and possibly extended to more datasets.
- In practice, one would want to select the coefficient values for $s, t$ based on some validation split, as described in section 4.5. In this scenario (not reported in the experiments), it is unclear whether the results would be significantly better than the standard Laplacian encoding.

Moreover, the overall contribution is too small for ICLR. The idea is very simple, so it should be complemented by a solid theoretical analysis or by an extensive experimental validation.

---

[1] Ito et al. Learning Laplacian Positional Encodings for Heterophilous Graphs. AISTATS 2025.

**Questions:**

- Why are the experiments limited to node-level tasks? The LPE has been used also for graph-level tasks. Do you think you method could be useful also in that scenario?

---

### Official Review · Reviewer_uFh8 · 2025-10-30

**Soundness:** 3
**Presentation:** 3
**Contribution:** 3
**Rating:** 4
**Confidence:** 4

**Summary:**

Node features are critical but can be limited in graph neural networks (GNNs). To remedy this issue, this paper proposes to derive spectral Laplacian eigenvectors of a family of graph matrices as augmented node features.

**Strengths:**

1.	The motivation of the proposed method is clearly explained and valid.

2.	This paper generalizes the idea of utilizing spectral embeddings in node feature augmentation. Moreover, theoretical analyses and explanations are provided.

3.	Various types of experimental settings are adopted for better validation.

**Weaknesses:**

1.	Using spectral embedding as node features has been researched early, both in GNNs and Graph Transformers, such as Specformer[1]. Please compare with those methods for a more convincing and comprehensive understanding.

[1] Specformer: Spectral Graph Neural Networks Meet Transformers, ICLR 2023

2.	This paper provides a clear theoretical understanding of the proposed method. However, theoretical power or performance is not discussed. Lemma 3.1 is straightforward.

3.	Calculating eigenvectors involves high computation overhead, especially in real-world networks. Please provide an analysis or empirical results on how the proposed method handles the efficiency issue.

4.	There are some explicit typos in the paper, such as “otherwise” in line 113.

**Questions:**

Please refer to my weakness comments, especially W2 and W3.

---

### Official Review · Reviewer_VLFB · 2025-10-31

**Soundness:** 2
**Presentation:** 1
**Contribution:** 2
**Rating:** 2
**Confidence:** 4

**Summary:**

This paper addresses the challenge of applying Graph Neural Networks (GNNs) to graphs with limited or absent node features. The authors propose a feature augmentation technique called Interpolated Laplacian Embeddings (ILEs). These embeddings are derived as the eigenvectors of a two-parameter family of graph matrices, $M(t,s) \equiv tD - sA$, which generalizes the standard graph Laplacian ($t=1, s=1$). The authors provide a theoretical interpretation of this matrix family using quadratic forms, arguing that the parameters $(t, s)$ allow the resulting embeddings to interpolate between capturing community structure and core-periphery structure. Empirical results on synthetic and real-world node classification tasks suggest that, for certain choices of $(t, s)$, ILEs can outperform standard Laplacian embeddings as augmented features.

**Strengths:**

S1. The paper tackles a practical and significant problem. The reliance of GNNs on high-quality node features is a well-known limitation.

S2. The theoretical analysis in Section 3.2 provides an intuitive interpretation of the $M(t,s)$ family. By analyzing the quadratic form $x^{\top}M(t,s)x$, the authors clearly explain how the parameters $t$ and $s$ balance the Laplacian quadratic form (favoring community structure) and the adjacency quadratic form (favoring core-periphery structure).

S3. The experimental evaluation tests the proposed method on synthetic Stochastic Block Models (SBMs), real-world datasets without any node features, and real-world datasets with node features corrupted by noise.

**Weaknesses:**

W1. The central idea of using a generalized graph matrix is not new. The proposed $M(t,s) = tD - sA$ family is a simplified version of the "universal adjacency matrices" family and is closely related to other existing families, such as the "deformed Laplacian", which the paper itself discusses. The proof in Lemma 3.1, showing that ILEs subsume the deformed Laplacian (up to an identity shift), is straightforward.

W2. A significant weakness is the absence of a method to select the $s$ and $t$ parameters. The experiments demonstrate that some $(s, t)$ pair from a manually selected grid can outperform the Laplacian baseline. This finding is expected; given a two-parameter family, it is highly probable that some combination will outperform a fixed baseline ($s=1, t=1$) on a given dataset. The paper's contribution would be much stronger if it proposed a data-driven method to select or learn $s$ and $t$, or at least provided a deeper analysis linking specific graph properties to optimal $(s, t)$ values. The suggestions in Section 4.5 (e.g., "perform cross-validation") are generic and computationally expensive, as this would require repeated eigendecompositions.

W3. The experiments are missing key baselines. The primary comparison is against the standard Laplacian and the adjacency matrix. Given the discussion of other matrices like the other Laplacian and the deformed Laplacian, it is unclear why these were not included as direct baselines. More importantly, the paper positions itself within the spectral augmentation literature but fails to compare ILEs against any other spectral augmentation or positional encoding methods.

W4. The paper's organization and writing is not good. This extensive background, which includes theorems and definitions from other papers, could be streamlined to focus more on the paper's specific contributions.

**Questions:**

Q1. The caption for Figure 1 mentions "Laplacian embedding ($k=1$)" and "Adjacency embedding ($k=1$)." What does $k$ represent?

Q2. The paper suggests cross-validation for $s$ and $t$, which seems computationally prohibitive, requiring a new eigendecomposition for each $(s, t)$ pair. Have the authors studied the sensitivity of performance to $s$ and $t$?


Q3. The quadratic form interpretation in Section 3.2 is key to the paper's argument. How much of this analysis is new versus a direct application of existing analysis from work on related matrices, like the deformed Laplacian?

---

### Official Review · Reviewer_tWch · 2025-10-31

**Soundness:** 3
**Presentation:** 3
**Contribution:** 2
**Rating:** 2
**Confidence:** 3

**Summary:**

In this paper, the authors propose using eigenvectors of a family of interpolating matrices — which include the Adjacency and Laplacian matrices as particular cases — for positional encoding in GNNs. The interpolation depends on two parameters, and the authors report results for these different augmentations on node classification tasks.

**Strengths:**

The paper is generally well written. Although there are several works on spectral positional encodings for GNNs, the specific experiments on how these embeddings behave when using interpolating matrices appear to be new.

**Weaknesses:**

I believe the paper falls short. The idea is interesting, but the fact that eigenvectors from different matrices have different effects on downstream tasks is already known (as the authors themselves acknowledge), and the experiments presented here merely confirm this. There is no real new insight into when one should prefer one matrix over another — or, equivalently, one set of (t,s) parameters over another. In this sense, I think that the contribution is incremental.

**Questions:**

Although the matrix M(t,s) in Equation (2) depends on two parameters, the eigenvectors depend (up to a constant) only on the ratio of these parameters. Therefore, it seems that the whole positional encoding scheme effectively depends on a single parameter. Please verify if this is indeed the case.

Finally, I find the citation to Spielman for the "Courant–Fischer theorem" on page 5 inappropriate. Citing a 2019 paper for a result established over a century ago seems unusual.

---

### Author Response · Authors · 2025-12-03
**Reply to Reviews**

We thank the reviewers for the review and comments. We do not provide a revised manuscript here (due to updated ICLR policy this year that do not allow for reviewer score updates), but we instead outline a plan for future revisions in response to reviewer comments below.

We appreciate the reviewers in recognizing that our paper is well written/explained clearly (Reviewer tWch, Reviewer uFh8, Reviewer NDbB, Reviewer VLFB) and that the problem we tackle is important/interesting (Reviewer tWch, Reviewer VLFB). We remark that we view a simple method that works as a virtue rather than a vice, so we appreicate that the simplicity/light-weight nature of our proposed method is recognized (Reviewer NDbB, Review A9i6).

We thank reviewers for suggesting further references (that we will include) and pointing out typos (that we will fix) in future revisions of the paper.

We want to briefly acknowledge and reply to certain core aspects of the reviewers' comments:

1. How to choose amongst matrices in the propose family (i.e. choosing over s, t parameters) is a point that is raised by multiple reviewers. In particular, Review Ai96 raised the point of considering more structure–label linkage evidence, which we appreciate. We mentioned cross validation as a potential method in the paper, but in future revisions we will explicitly propose a statistically principled method for selecting s, t parameters that take into account label structure

2. Several reviewers recommend comparing to more baselines. We note here that

-  certain basellines such as the Adjacency matrix/Laplacian/deformed Laplacian matrix augmentations are special cases of our family, so these baselines are already incorporated in our experiments. Reviewers might have reached the mistaken conclusion that "we did not compare any baselines" since we did not *explicitly label* these special cases in our tables. To avoid such confusion, we will explicitly label these special cases in our tables in future revisions. We do appreciate Reviewer NDdB's suggestion on comparing with concatenated embeddings, which we will incorporate in future revisions.

- We did incorporate SignNets in our experiments (we will make this more explicit in future revisions). We will provide further discussion/citations to related work in our future revisions.

- Moreover, we evaluated our methods on over 10 real world datasets. Due to space limits, some of these results are included in the appendix rather than in the main text. In future revisions, we will restructure our text (as suggested by some reviewers) so that more of these results can appear in the main text. As such, we find certain comments that we used "few datasets" to be mistaken.

- We will investigate the references that reviewers suggested and, where appropriate, add them as baselines in our experimental evaluation. Note that our focus is on the node classification context since it is one of the most prevalent benchmark tasks that can serve as testbed for new ideas. We do not make any claims in the graph transformer case, since it is an entirely different problem with very different architectures as the ones considered in our paper. We will continue to cite work in the transformers literature, but we view it as mostly distracting to consider comparison against transformers when our claims and focus are on the node classification setting.

3.  We emphasize that the goal of this paper is not to propose any new spectral graph theory. Nor do we claim that we are the first to propose generalized graph matrices (in fact, we explicitly cite many such prior work). Rather, the novelty of our ideas and contribution is in the *application of these insights in the graph neural networks spectral augmentation setting*, which often only considers simple spectral embeddings (like the Laplacian) or one-hot vectors. Note that there are other position encoding methods out there that are much more complicated/depend on learned encodings. The advantage of spectral augmentation lies in its simplicity to provide prior/structural information, without needing to learn such embeddings. We aim to make this point more clear in our future revisions.

---

### Meta-Review · Area_Chair_xvsL · 2026-01-01

**Summary:**

No response was provided during the rebuttal phase. As a result, the reviewers’ concerns remain unaddressed, and the paper is not ready for publication in its current form. I encourage the authors to consider these comments carefully, as addressing them could significantly strengthen a revised version of the work.

**Reviewer Concerns:**

No rebuttal was provided, and therefore, the reviewers’ concerns remain unaddressed.

**Reviewer Scores:**

No score changes were made, as no rebuttal was provided.

---

### Decision · Program_Chairs · 2026-01-26

Reject